# Circulating Levels of the Soluble Receptor for AGE (sRAGE) during Escalating Oral Glucose Dosages and Corresponding Isoglycaemic i.v. Glucose Infusions in Individuals with and without Type 2 Diabetes

**DOI:** 10.3390/nu12102928

**Published:** 2020-09-24

**Authors:** Amelia K. Fotheringham, Jonatan I. Bagger, Danielle J. Borg, Domenica A. McCarthy, Jens J. Holst, Tina Vilsbøll, Filip K. Knop, Josephine M. Forbes

**Affiliations:** 1Mater Research Institute, The University of Queensland, Translational Research Institute, Woolloongabba, QLD 4102, Australia; amelia.fotheringham@mater.uq.edu.au (A.K.F.); danielle.borg@mater.uq.edu.au (D.J.B.); domenica.mccarthy@mater.uq.edu.au (D.A.M.); 2School of Biomedical Sciences, Faculty of Medicine, The University of Queensland, St Lucia, QLD 4072, Australia; 3Center for Clinical Metabolic Research, Herlev-Gentofte Hospital, University of Copenhagen, 2900 Hellerup, Denmark; JIBagger@dadlnet.dk (J.I.B.); t.vilsboll@dadlnet.dk (T.V.); filip.krag.knop.01@regionh.dk (F.K.K.); 4Novo Nordisk Foundation Center for Basic Metabolic Research, Faculty of Health and Medical Sciences, University of Copenhagen, 2200 Copenhagen, Denmark; jjholst@sund.ku.dk; 5Department of Biomedical Sciences, Faculty of Health and Medical Sciences, University of Copenhagen, 2200 Copenhagen, Denmark; 6Department of Clinical Medicine, Faculty of Health and Medical Sciences, University of Copenhagen, 2200 Copenhagen, Denmark; 7Steno Diabetes Center Copenhagen, 2820 Gentofte, Denmark; 8Mater Clinical School, Faculty of Medicine, The University of Queensland, South Brisbane, QLD 4101, Australia

**Keywords:** RAGE, sRAGE, AGEs, type 2 diabetes mellitus, diabetes complications, metabolic alterations of T2DM, enteroendocrine

## Abstract

Postprandial glucose excursions are postulated to increase the risk for diabetes complications via the production of advanced glycation end products (AGEs). The soluble receptor of AGEs (sRAGE) likely acts as a decoy receptor, mopping up AGEs, diminishing their capacity for pro-inflammatory and pro-apoptotic signaling. Recent evidence suggests that AGEs and soluble receptor for AGEs (sRAGE) may be altered under postprandial and fasting conditions. Here, we investigated the effects of increasing oral glucose loads during oral glucose tolerance tests (OGTT) and matched isoglycaemic intravenous (i.v.) glucose infusions (IIGI) on circulating concentrations of sRAGE. Samples from eight individuals with type 2 diabetes and eight age-, gender-, and body mass index (BMI)-matched controls, all of whom underwent three differently dosed OGTTs (25 g, 75 g, and 125 g), and three matched IIGIs were utilised (NCT00529048). Serum concentrations of sRAGE were measured over 240 min during each test. For individuals with diabetes, sRAGE area under the curve (AUC_0–240_
_min_) declined with increasing i.v. glucose dosages (*p* < 0.0001 for trend) and was lower during IIGI compared to OGTT at the 125 g dosage (*p* = 0.004). In control subjects, sRAGE AUC_0–240_
_min_ was only lower during IIGI compared to OGTT at the 25 g dose (*p* = 0.0015). sRAGE AUC_0–240_
_min_ was negatively correlated to AUC_0–240_
_min_ for the incretin hormone glucagon-like peptide −1 (GLP-1) during the 75 g OGTT and matched IIGI, but only in individuals with type 2 diabetes. These data suggest that gastrointestinal factors may play a role in regulating sRAGE concentrations during postprandial glucose excursions, thus warranting further investigation.

## 1. Introduction

Variability in plasma glucose concentrations has gained increasing attention as a risk factor for diabetes complications, including kidney disease. The mechanisms by which glucose variability may mediate end-organ complications, however, are not fully understood. Advanced glycation end products (AGEs) are a heterogeneous group of post-translational modifications, produced following the binding and rearrangement of reducing sugars. AGEs are formed at accelerated rates in response to chronic hyperglycaemia, hyperlipidaemia, and oxidative stress [1]. Increases in circulating concentrations of AGEs, such as carboxymethyllysine (CML) [2,3,4], and their glycation precursors, such as haemoglobin A_1c_ (HbA_1c_) [5], predict cardiovascular mortality in individuals with both type 2 [2,3], and type 1 diabetes [4].

Many AGEs, including CML, interact with the receptor for AGEs (RAGE), a full-length membrane bound isoform, to elicit signalling pathways facilitating inflammation, autophagy, proliferation, and apoptosis [6]. Soluble isoforms of RAGE have also been characterised and are thought to act as competitive decoys, diminishing the signalling capacity of membrane bound RAGE [6]. These include soluble RAGE (sRAGE), which is produced following cleavage of the membrane bound isoform by metalloproteinases (C-truncated RAGE) or through alternative splicing as endogenous secretory RAGE (esRAGE), which is actively secreted [6]. Prospective cohort studies have found that elevations in circulating sRAGE are associated with a reduced incidence of both cardiovascular disease and all-cause mortality in the context of both type 1 diabetes and type 2 diabetes [4,7,8].

Gastrointestinal factors meticulously control circulating glucose concentrations by stimulating appropriate insulin secretion via the actions of the incretin hormones glucagon-like peptide 1 (GLP-1) and glucose-dependent insulinotropic peptide (GIP) [9]. These two incretin hormones are responsible for up to 60% of insulin secretion in response to a carbohydrate-rich meal (the incretin effect) [10]. In type 2 diabetes, the incretin effect is impaired, thereby exacerbating hyperglycaemia [11,12]. Recent evidence suggests that circulating sRAGE is dynamically regulated between the fasted and the fed state in patients with diabetes [13]. Furthermore, hormones produced by the gastrointestinal system may regulate cellular expression of RAGE [14,15].

It is not clear from any study to date whether glucose excursions, as would occur postprandially, dynamically alter systemic sRAGE concentrations, nor the impact of gastrointestinal factors on RAGE regulation including its cellular expression [15,16]. Hence, the objective of the present study was to assess the temporal dynamics of the receptor sRAGE in response to increasing glucose loads. Glucose was administered either orally or by isoglycaemic i.v. glucose infusions (IIGIs), which bypassed the gastrointestinal factors involved in the regulation of glucose, such as the incretin hormones. Further to this, this study aimed to determine if receptor dynamics are altered by diabetes, where significantly greater glucose variability occurs and enteroendocrine signalling is impaired.

## 2. Materials and Methods

### 2.1. Study Participants

Analysis of serum concentrations of sRAGE in a cohort of individuals with well controlled type 2 diabetes (*n* = 8) and matched healthy controls (*n* = 8) was performed [11]. The characteristics of recruited participants have been reported in detail elsewhere [11]. The participants with type 2 diabetes were diagnosed according to the criteria of the World Health Organisation (WHO) [11], with an average duration of diabetes of 8 (range: 6–36) months. These subjects were treated with diet and exercise only. Healthy control subjects were matched for sex, body mass index (BMI; + −3 kg/m^2^) and age (+ −5 years), were without family history of diabetes and had normal glucose tolerance, ascertained by a 75 g-OGTT performed immediately prior to inclusion in the study. None of the participants had impaired renal function or microalbuminuria, proliferative retinopathy or impaired liver function and were negative for islet cell and glutamic acid decarboxylase (GAD65) autoantibodies. No medications thought to influence glucose, insulin, C-peptide, or incretin hormone responses were used by participants. This study was completed in accordance with the Declaration of Helsinki, following ethical approval from the Scientific-Ethics Committee of the Capital Region of Denmark (H-A-2007-0048) and registered at clinicaltrials.gov (NCT00529048). All participants consented to participate after receiving oral and written information.

### 2.2. Experimental Procedures

Each participant underwent three separate OGTTs using increasing glucose loads 25 g, 75 g and 125 g, in randomised order on multiple occasions. The participants were studied in the morning in a recumbent position after an overnight (10 h) fast. Blood samples were drawn from a catheter inserted into the cubital vein for the OGTT. Three paired IIGIs of the same duration were then performed, during which plasma glucose concentrations were matched to each OGTT load. IIGIs were performed in randomised order under the same conditions as the OGTTs. On IIGI days, cannulas were inserted into cubital veins of both arms for collection of arterialised blood samples (as per OGTT time points) and glucose infusion. Glucose infusion rate was adjusted, and reproduced plasma glucose concentrations obtained during the matched OGTT as previously described [16]. Serum prepared from blood collected at time points 0, 10, 30, 60, 120, 240 min was immediately stored at −80 °C until subsequent analysis. At the first thaw, multiple aliquots were made of each sample to avoid repeated freeze thaw cycles.

### 2.3. Assessment of Serum RAGE Concentrations

Circulating sRAGE was assayed using a solid-phase sandwich ELISA (Duo Set Human RAGE ELISA, R&D Systems, Minneapolis, United States of America) according to the manufacturer’s instructions and detected using a Pherastar FS microplate reader (BMG Labtech, Germany). Inter-plate %CV was 10.7%. Baseline fasting sRAGE was calculated as the mean baseline fasting concentration at time 0 across the 6 experimental procedures for each participant. All samples were analysed by ELISA within a two-week period.

### 2.4. Statistical Methods

The sample size was calculated previously to detect changes in incretin hormones as described by Bagger et al. [11]. Statistical analysis was performed using GraphPad Prism, version 8.0.1 (San Diego, United States of America). Pearson’s or Spearman’s univariate analyses was used to examine relationships among patient characteristics as indicated. Area under the curve (AUC) values were calculated using the trapezoidal rule. Comparison of AUC between OGTT and matched IIGI was performed by paired T test (two tailed) or non-parametric equivalent (Wilcoxon matched pairs signed rank test) while comparison between controls and patients with diabetes was performed by student’s two tailed T test. A comparison of the three dosages (dose effect) was performed using a repeated measures one-way ANOVA (parametric) or Friedman’s test (non-parametric) as indicated, followed by the test for linear trend where significance was reached. Comparison of curves (all time points) between OGTT and IIGI at each dosage was performed by mixed models with participant ID as random factor.

## 3. Results

### 3.1. Glucose Homeostasis during OGTT and IIGI

Glucose, insulin, C-peptide, GIP, GLP-1, and glucagon in participants during OGTT and IIGI have been previously reported [11,17]. Additionally, gastric emptying was reported for OGTT [11]. Briefly, plasma glucose concentrations increased with escalating oral glucose loads in individuals with type 2 diabetes due to impaired insulin secretion and incretin responses when compared to matched control individuals. The suppression of glucagon secretion following a glucose bolus was also blunted in individuals with type 2 diabetes [17].

### 3.2. At baseline, sRAGE Concentrations were not Related to Anthropometric Characteristics or Measures of Glucose Homeostasis

By design, participants with type 2 diabetes had higher fasting plasma glucose concentrations, higher peak stimulated C-peptide and HbA_1c_ at baseline but did not differ from the control group in bodyweight, BMI, waist circumference or age. Baseline, fasting sRAGE concentrations did not associate with participant characteristics such as age, BMI, or plasma glucose in either group. Given participants had only short duration of diabetes (group mean <1 year, Table 1), we also examined all participants as a single cohort. When all participants were combined as a single cohort baseline fasting serum sRAGE concentrations tended to be negatively associated with waist circumference (*p* = 0.08, Table 1).

### 3.3. Circulating sRAGE Concentrations Following Glucose Administration

When all participants were analysed as a single group, AUC_sRAGE 0–240 min_ was lower during IIGI compared to OGTT at the 125 g glucose load (Table 2, Figure 1). No dose effects were seen for AUC_sRAGE 0–240 min_ during OGTT or IIGI (Table 2). Comparisons of OGTT and IIGI curves by mixed models at each glucose dose, showed significant effects of time post-dose, as well as interactive effects between time and administration route at all dosages (Figure 1).

When participants with diabetes and controls were analysed separately, significant main effects of time and interaction of time and administration were present at all dosages in participants with type 2 diabetes (Figure 1B). In contrast, significant effects were only seen at the 25 g glucose dose in controls (Figure 1C). Participants with type 2 diabetes showed a decrease in AUC_sRAGE 0–240 min_ during IIGI compared to the OGTT at the 125 g glucose loads (Figure 1B), but no differences at lower dosages. In control participants, AUC_sRAGE 0–240 min_ was lower during IIGI as compared with OGTT only at the 25 g dose (Figure 1C). There were no glucose dosage effects on sRAGE seen in control participants for orally or intravenously administered glucose (Table 2). However, in individuals with type 2 diabetes, there was a significant dose effect of intravenous glucose on circulating sRAGE during IIGI (*p* = 0.0033, Table 2), with AUC_sRAGE 0–240 min_ modestly decreasing as glucose load increased (*p* for trend <0.001, r^2^ = 0.045). Univariate analyses of the associations between AUC_sRAGE 0–240_ and AUC_0–240_ for glucose and the glucose regulating hormones insulin, glucagon, GLP-1, and GIP showed strong negative relationships between sRAGE and GLP-1 in individuals with type 2 diabetes, both during OGTT and IIGI at the 75 g dose (Table 3). 

## 4. Discussion

In the present study, we used biobanked serum samples to examine whether circulating levels of the receptor sRAGE, were increased or modified in response to increasing glucose loads in individuals with and without type 2 diabetes. Additionally, we examined whether gastrointestinal factors and portal delivery of glucose were important for circulating sRAGE responses. Here, we report that circulating sRAGE concentrations were modestly decreased in response to intravenously administered glucose compared to oral glucose at higher dosages in participants with type 2 diabetes. Furthermore, sRAGE (AUC_0–240_) showed strong negative correlation with the incretin hormone GLP-1 in individuals with type 2 diabetes, both during OGTT and IIGI at the 75 g dose (Table 3), which could suggest a diabetes specific gastrointestinal component for sRAGE regulation that warrants further investigation.

### 4.1. Baseline Correlations

In contrast to previous studies comparing circulating sRAGE in individuals with type 2 diabetes and controls [18,19], no differences in baseline levels of sRAGE were identified between the control participants and patients with type 2 diabetes. This may reflect the short duration of diabetes (<1 year from diagnosis) of participants in this study compared to previous studies [18,19]. Further, factors previously reported to influence circulating levels of sRAGE and sRAGE ligands such as AGEs [20,21] were stringently matched between groups in the present study, including age, adiposity, and gender. Baseline sRAGE serum concentrations tended to be negatively associated with waist circumference, in agreement with previous studies showing lower circulating sRAGE [22,23] and AGE [24,25] concentrations in obese or overweight individuals populations as compared to lean controls. This decline in circulating sRAGE with obesity is postulated to result from sequestration of AGEs by sRAGE into adipose tissue, as reported in pre-clinical models [26]. Here, we report no association between baseline sRAGE and measures of glucose homeostasis but this likely reflects that participants only had very recently diagnosed type 2 diabetes and the narrow BMI range represented by our participants due to the stringent matching of those individuals with type 2 diabetes and controls.

### 4.2. Temporal and Net Changes in sRAGE Differ Between Oral Ingestion and Infusion of Glucose

The production of circulating sRAGE occurs through the proteolytic cleavage of membrane bound RAGE and alternative splicing during gene expression [6]. sRAGE serves as a decoy receptor mediating the interaction of RAGE ligands such as CML with the membrane bound RAGE orchestrating cellular signalling and function [6]. sRAGE levels can be dynamically and rapidly upregulated in acute settings, such as trauma [27] and myocardial infarction [28]. Interestingly, trauma also commonly results in acute hyperglycaemia concomitantly with increases in sRAGE [27,29,30,31], and AGE precursors such as reactive dicarbonyls [30,31,32,33]. To our knowledge, however, this is the first time that circulating levels of sRAGE have been studied in the context following OGTTs and matched IIGIs in healthy individuals or patients with type 2 diabetes.

Changes in sRAGE did not appear proportional to glucose dose during either the OGTT or the IIGI in controls when analysed separately, although a modest decrease in sRAGE with increasing glucose infusion dosages was seen in individuals with diabetes. AUC _sRAGE 0–240 min_ was also lower during IIGI compared to OGTT in persons with type 2 diabetes, at the higher glucose dosages and this decrease was also noted at the lowest glucose dose, in the control cohort. A recent study subjected obese but otherwise healthy individuals to a 24 h hyperglycaemic clamp, reporting no significant changes in sRAGE at 2 h or 24 h of hyperglycaemia at 5.4 mmol/L above baseline [34]. In contrast, post-hoc analysis of our data found that, at 2 h, sRAGE was statistically below baseline (fasted) levels following the lowest and highest glucose infusions, in patients with type 2 diabetes. Given that glucose is often steadily increased during clamp infusions, this suggests that rapid, acute changes in blood glucose as simulated by the OGTT and IIGI, have greater acute effects on sRAGE than more gradual sustained glucose changes.

In the present study, serum sRAGE concentrations declined acutely following i.v. glucose infusion, but were less influenced by oral glucose dosage. Although we measured total soluble isoforms of RAGE, esRAGE, one isoform of sRAGE, increases significantly with short term caloric restriction in healthy individuals [35], whilst both total sRAGE and esRAGE increase acutely with fasting [13]. This suggests that, gastrointestinal factors may play a key role in the maintenance of circulating sRAGE concentrations during fluctuations in circulating glucose, glucagon, and insulin. The reasons for this gastrointestinal regulation of sRAGE remain unknown, suffice to say that RAGE is highly expressed by mucosal epithelia where it plays a role in host-pathogen defence [36]. AUC_0–240_ for the gut hormone, GLP-1 and sRAGE were significantly correlated during the 75 g OGTT and the matched IIGI subjects with type 2 diabetes. Although correlative data should be interpreted cautiously, GLP-1 agonists have previously been shown to attenuate AGE induced RAGE expression in a small number of in-vivo and in-vitro settings [14,15], leading several researchers to postulate that the GLP-1-AGE-RAGE axis may be important in diabetes complications [37,38] and obesity-related metabolic asthma [39]. Taken together, this suggests a potential diabetes specific role for GLP-1 in sRAGE homeostasis that warrants future investigation.

The overall balance of sRAGE to its ligands may be more critical than the individual concentrations. Only a limited number of studies have directly examined the AGE:sRAGE ratio in the context of diabetes or diabetes complications [40,41]. However, a randomised crossover study of dietary AGE intake in healthy overweight individuals, did not show any changes in sRAGE with either low or high dietary AGE consumption [42], suggesting that circulating sRAGE is not proportional to circulating AGEs in this setting. This is the opposite to what occurs during situations of trauma and myocardial infarction where stress-induced hyperglycaemia, AGEs and sRAGE are all reported to increase [27,29,30,31,32,33]. This implies that circulating AGE and sRAGE changes are most consistently seen in response to stress-induced gluconeogenesis, where inflammation is concomitantly observed. This warrants following up in future studies given the prominent role of sRAGE in inflammatory processes.

### 4.3. Limitations

This study was statistically powered to detect differences in incretin hormones and glucagon [11]. As evidenced, there are considerably smaller effects sizes seen in the present study for sRAGE in response to increasing glucose loads increasing the risk of a type II statistical error. Although the repeated measures design and pairing of oral and i.v. glucose loads for every participant significantly increased the statistical power, these findings need to be strengthened in larger cohorts. Additionally, the extended storage time of the samples prior to analysis must be acknowledged, although serum RAGE has shown good stability during storage in previous analyses [43].

## 5. Conclusions

In the present study, sRAGE, as assessed by AUC_0–240 min_, showed few changes between individuals with type 2 diabetes and age-, gender-, and BMI-matched controls at equivalent dosages and administration routes. However, in individuals with diabetes, sRAGE measured as AUC_sRAGE 0–120 min_ decreased as i.v. glucose doses increased. This was not seen during the OGTTs, nor was it seen in control individuals and this suggests an as yet undiscovered link between the gastrointestinal regulation of postprandial hormonal response and acute changes in sRAGE in diabetes.

## Figures and Tables

**Figure 1 nutrients-12-02928-f001:**
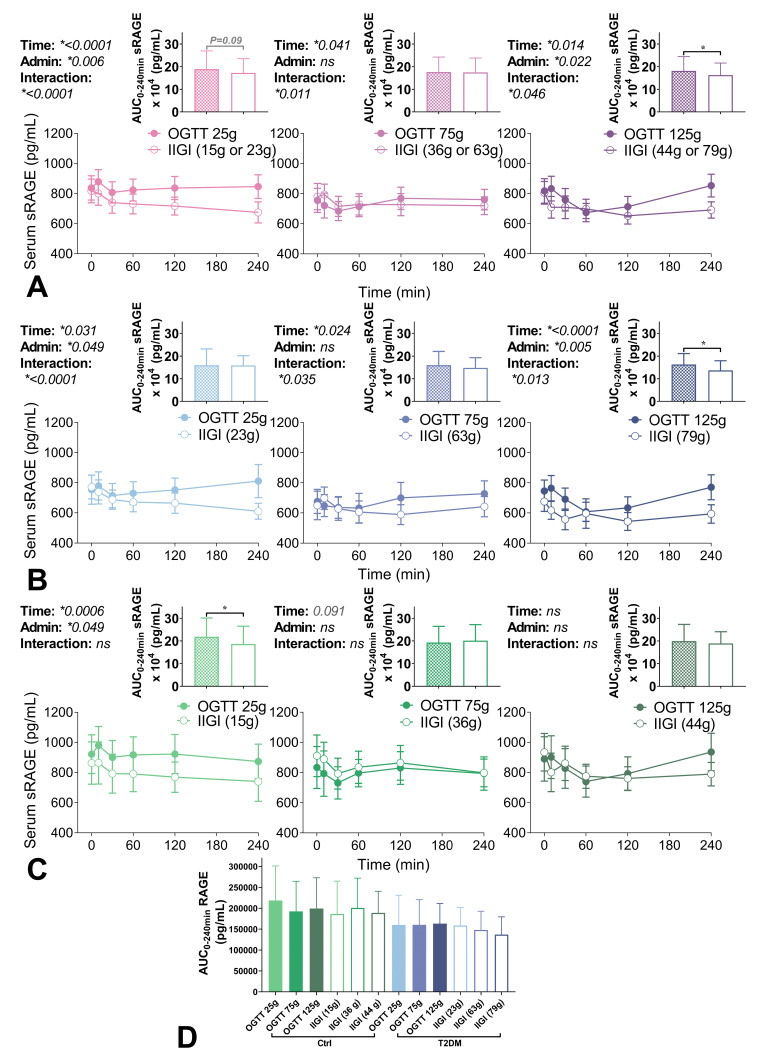
Circulating soluble receptor for advanced glycation end products (sRAGE) concentrations during paired oral glucose tolerance tests (OGTT) and isoglycaemic intravenous glucose infusions (IIGI) at different glucose dosages. Three OGTTs (25 g, 75 g or 125 g; filled circles/bars) and three matched IIGI (open circles/bars) were performed over 240 min (min) in (**A**) all participants (*n* = 16, purple), (**B**) participants with recently diagnosed type 2 diabetes (*n* = 8, blue) and (**C**) matched controls (*n* = 8, green). Curves show mean ± SEM values for sRAGE across time and were analysed by mixed model. *p* values for individual effects of time, administration (admin; oral vs intravenous) and interaction terms are shown above (left) each curve (* *p* < 0.05; *ns* denotes no significant difference). Area under the curve (AUC) was calculated for curves and presented above as bar graphs showing mean ± SD. AUC were analysed by T test (* *p* < 0.05). (**D**) shows all AUC_sRAGE 0–240_ min for controls (green) and individuals with type 2 diabetes (blue) for all glucose doses and administration routes (OGTT filled bars, IIGI open bars).

**Table 1 nutrients-12-02928-t001:** Study participant anthropometric characteristics and their relationship with circulating sRAGE at baseline.

	No Diabetes (Control) *n* = 8	Type 2 Diabetes *n* = 8	*p* Value	Baseline Associations to sRAGE, *r, p*		
				Control ^a^	Type 2 Diabetes ^a^	All ^b^
Sex (M:F)	3:5	3:5	*ns*	-	-	-
Age (years)	56.5 ± 11.4	57.0 ± 11.9	*ns*	0.124 *ns*	−0.050 *ns*	0.025 *ns*
BMI (kg/m^2^)	28.9 ± 2.1	29.5 ± 3.2	*ns*	−0.212 *ns*	−0.024 *ns*	−0.150 *ns*
Waist circumference (cm)	101.8 ± 12.40	105.0 ± 10.18	*ns*	−0.425 *ns*	−0.3631 *ns*	−0.451 *0.08*
Systolic Blood Pressure (mmHg)	125.9 ± 14.08	131.4 ± 14.73	*ns*	0.078 *ns*	0.240 *ns*	0.089 *ns*
HbA_1C_ (%)	5.4 ± 0.3	7.0 ± 0.7	*<0.001*	−0.488 *ns*	0.266 *ns*	−0.293 *ns*
Duration of diabetes (months)	-	8.4 ± 11.8	-	-	0.609 *ns*	-
Plasma glucose t = 120 mins (mmol/L)	6.0 ± 0.6	16.0 ± 2.7	*<0.001*	0.455 *ns*	−0.496 *ns*	−0.147 *ns*
Fasting plasma insulin (mmol/L)	56.42 ± 20.93	86.34 ± 39.51	*ns*	−0.072 *ns*	−0.207 *ns*	−0.260 *ns*
Peak C-peptide (75 g OGTT)	3.839 ± 0.90	2.798 ± 0.93	*0.039*	−0.391 *ns*	−0.188 *ns*	−0.100 *ns*
Fasting serum sRAGE (pg/mL)	892.2 ± 371.1	712.0 ± 222.1	*ns*	-	-	-

Anthropometric data are shown as mean ± SD. Pearson ^a^ or spearman ^b^ coefficients of correlation are presented and *p* values (italicised) are shown (*ns*- non-significant). Abbreviations: BMI-body mass index, HbA_1C_-haemoglobin A1c, OGTT-oral glucose tolerance test, sRAGE—soluble receptor of advanced glycation end-products, All—all subjects combined as a single cohort.

**Table 2 nutrients-12-02928-t002:** Comparison of AUC_0–240 min_ for sRAGE during 25 g, 75 g and 125 g-OGTTs and corresponding IIGIs.

	AUC_0–240_ sRAGE (ng/mL)			
	Controls	Type 2 Diabetes	*p* Value	All
25 g OGTT	218.44 ± 82.79	159.92 ± 71.18	*ns*	181.89 ± 80.47
Matched IIGI	186.29 ± 78.61	158.73 ± 42.90	*ns*	172.81 ± 62.81
75 g OGTT	192.50 ± 71.90	160.26 ± 60.27	*ns*	176.38 ± 66.22
Matched IIGI	200.99 ± 71.07	148.03 ± 44.94	*0.09*	174.51 ± 63.62
125 g OGTT	199.34 ± 73.68	163.10 ± 48.29	*ns*	181.22 ± 63.02
Matched IIGI	188.85 ± 51.84	136.99 ± 42.45	** 0.046*	162.92 ± 53.03
Intra-group variation OGTT (*p* value)	*ns*	*ns*		*ns*
Intra-group variation IIGI (*p* value)	*ns*	** 0.0033* *p_(trend)_ < 0.001* *r^2^ = 0.045*		*ns*

AUC_0–240 min_ is shown as mean ± SD. *p* values are. * *p* < 0.05 following comparison by T test (*ns*—non-significant). AUC_0–240 min_ was compared for the three dosages to determine a dose effect (intra-group variation) by one-way rmANOVA followed by post-test for linear trend (low-high glucose). Abbreviations: IIGI- isoglycaemic intravenous glucose infusion, OGTT-oral glucose tolerance test, sRAGE—soluble receptor of advanced glycation end-products, All—all subjects combined as a single cohort.

**Table 3 nutrients-12-02928-t003:** Univariate relationships between AUC_0–240 min_ for sRAGE with AUC_0–240_ for glucose, insulin glucagon, and incretin hormones GLP-1 and GIP.

	Controls					Type 2 Diabetes					All				
	Glucose	Insulin	Glucagon	GLP1	GIP	Glucose	Insulin	Glucagon	GLP1	GIP	Glucose	Insulin	Glucagon	GLP1	GIP
25 g OGTT	0.095 *ns ^b^*	0.143 *ns ^b^*	0.095 *ns ^b^*	−0.071 *ns ^b^*	0.524 *ns ^b^*	−0.338 *ns ^a^*	0.088 *ns ^a^*	−0.085 *ns ^a^*	−0.604 *ns ^a^*	0.334 *ns ^a^*	−0.359 *ns ^b^*	0.15 *ns ^b^*	−0.233 *ns ^b^*	−0.444 *0.087 ^b^*	0.365 *ns ^b^*
Matched IIGI	0.503 *ns ^b^*	0.214 *ns ^b^*	−0.119 *ns ^b^*	−0.047 *ns ^b^*	0.539 *ns ^b^*	−0.388 *ns ^a^*	−0.530 *ns ^a^*	−0.224 *ns ^a^*	−0.705 *0.051 ^a^*	−0.0126 *ns ^a^*	−0.003 *ns ^b^*	−0.227 *ns ^b^*	−0.191 *ns ^b^*	−0.327 *ns ^b^*	0.225 *ns ^b^*
75 g OGTT	0.65 *0.08 ^a^*	0.376 *ns ^a^*	0.17 *ns ^a^*	0.119 *ns ^a^*	0.205 *ns ^a^*	−0.143 *ns ^b^*	−0.262 *ns ^b^*	−0.238 *ns ^b^*	−0.881 ** 0.007 ^b^*	−0.095 *ns ^b^*	−0.112 *ns ^b^*	0.026 *ns ^b^*	−0.212 *ns ^b^*	−0.194 *ns ^b^*	0.076 *ns ^b^*
Matched IIGI	−0.024 *ns ^b^*	0.286 *ns ^b^*	−0.071 *ns ^b^*	−0.405 *ns ^b^*	0.619 *ns ^b^*	0.043 *ns ^a^*	−0.096 *ns ^a^*	0.038 *ns ^a^*	−0.795 ** 0.018 ^a^*	0.205 *ns ^a^*	−0.353 *ns ^b^*	−0.065 *ns ^b^*	−0.171 *ns ^b^*	−0.615 ** 0.013 ^b^*	0.285 *ns ^b^*
125 g OGTT	0.674 *ns ^a^*	0.403 *ns ^a^*	0.322 *ns ^a^*	−0.390 *ns ^a^*	0.204 *ns ^a^*	−0.238 *ns ^b^*	−0.381 *ns ^b^*	−0.524 *ns ^b^*	−0.310 *ns ^b^*	0.167 *ns ^b^*	−0.241 *ns ^b^*	0.082 *ns ^b^*	−0.365 *ns ^b^*	−0.335 *ns ^b^*	0.165 *ns ^b^*
Matched IIGI	0.238 *ns ^b^*	−0.024 *ns ^b^*	0.00 *ns ^b^*	−0.286 *ns ^b^*	0.238 *ns ^b^*	0.061 *ns ^a^*	−0.178 *ns ^a^*	−0.204 *ns ^a^*	−0.538 *ns ^a^*	0.249 *ns ^a^*	−0.356 *ns ^b^*	−0.185 *ns ^b^*	−0.268 *ns ^b^*	−0.371 *ns ^b^*	0.131 *ns ^b^*

Pearson’s ^a^ or Spearman’s ^b^ univariate correlations were performed after examining data distribution. Correlation coefficients are shown with *p* values (italicised). *p* values for significant correlations (* *p* < 0.05; dark grey) and those trending towards significance (0.05 < *p* > 0.1; light grey) are reported (*ns-* non-significant). Abbreviations: IIGI- isoglycaemic intravenous glucose infusion, OGTT-oral glucose tolerance test, sRAGE—soluble receptor of advanced glycation end-products, GLP-1—glucagon-like peptide 1, GIP, glucose-dependent insulinotropic peptide, All—all subjects combined as a single cohort.

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
