# Peer review of "Circulating Levels of the Soluble Receptor for AGE (sRAGE) during Escalating Oral Glucose Dosages and Corresponding Isoglycaemic i.v. Glucose Infusions in Individuals with and without Type 2 Diabetes"

_nutrients, 2020, doi:10.3390/nu12102928_

Round 1

Reviewer 1 Report

This manuscript reports on the results of a study conducted about 10 or more years ago that was originally designed, powered, and published to study the impact of the treatments on incretin hormone concentrations. In this manuscript the serum concentrations of carboxymethyllysine (CML) and the soluble receptor for advanced glycation endproducts (sRAGE) are reported on from this same study design.  However, the study was not powered for these outcome measures and the specifics of sample collection and timeframe of storage relative to analysis are not provided.  The variance reported was also quite high leading to a concern for the validity of the results for these two primary outcome measures. The manuscript is missing complete information about sample collection, storage and analyte stability.

Author Response

We would, like to thank the reviewer for their careful consideration of the manuscript and constructive feedback of how to improve this work. Our itemised responses are listed below.

Reviewer comment:

In this manuscript the serum concentrations of carboxymethyllysine (CML) and the soluble receptor for advanced glycation endproducts (sRAGE) are reported on from this same study design.  However, the study was not powered for these outcome measures and the specifics of sample collection and timeframe of storage relative to analysis are not provided.  The variance reported was also quite high leading to a concern for the validity of the results for these two primary outcome measures. The manuscript is missing complete information about sample collection, storage and analyte stability.

  1. Study Statistical Power: We acknowledge that this study was not specifically powered to measure AGEs and sRAGE and we have deliberately highlighted this in the manuscript (see lines 117-118 and 269-273) acknowledging the potential for a type 2 error. However, this is a unique cohort, with the same subjects undergoing matched multiple OGTT and isoglycaemic glucose infusions, which allowed for subjects to be compared with themselves maxamising statistical power. It also has uniquely matched oral and intravenous glucose delivery where the blood glucoses are precisely matched at each timepoint. Both the repeated measures design and the rigorous matching between the cohorts adds significant extra power. This also provides unprecedented ability to examine AGEs and sRAGE production after eating or following gluoconeogenic glucose production (intravenous delivery) akin to those seen physiologically and unpack the differences between these. As a consequence, we believe that we have found some interesting and significant findings.
  2. Sample collection and variance: The laboratory work for this manuscript was actually carried out in 2015/2016 as part of a Doctor of Philosophy project (PhD). Hence the samples were approximately 9 years post collection at the time of analysis. Delay of publishing of this work is due to the student’s requirements to complete other projects within the scope of that PhD as a matter of priority. All samples were meticulously collected and aliquoted at collection, stored at -80°C for the entirety of the study and further aliquoted at first thaw to avoid freeze-thaw effects on the sample for AGEs and RAGE. It has previously been shown (see manuscript (i) below) that CML in serum and plasma is quite stable over extended periods of time and across multiple freeze thaws if stored correctly. Similarly, sRAGE has also been measured successfully in samples stored for a similar period of time to the study we describe ; please refer to manuscript (ii) below. We have added further detail to the manuscript and ensured that the methods section is more transparent regarding collection and storage of the samples (see lines 105-107 and 113-114). With regard to the variance reported, the variance seen with sRAGE is close to the limits seen in manuscript (ii). However, given the large variation we report for the CML measurements and the fact that CML does not in fact add to the findings of the manuscript, we have taken the decision to remove these data.

(i) Hull, G.L.J., et al., Validation study to compare effects of processing protocols on measured N (ε)-(carboxymethyl)lysine and N (ε)-(carboxyethyl)lysine in blood. Journal of clinical biochemistry and nutrition, 2013. 53(3): p. 129-133.).

(ii) Thomas, M.C., et al., Relationship Between Levels of Advanced Glycation End Products and Their Soluble Receptor and Adverse Outcomes in Adults With Type 2 Diabetes. Diabetes Care, 2015. 38(10): p. 1891-1897.

Reviewer 2 Report

In the manuscript submitted by Fotheringham et al., the authors present interesting data regarding Advanced Glycation End Products (AGEs), as carboxymethyllysine, by offering some novel information related to their relevance in type 2 diabetes mellitus.
The Introduction and aims of their work are well described, and the study was designed accordingly, with an adequate Methodology. The presentation and interpretation of their Results are clearly detailed. All the Figures and Tables, as well as the Supplementary Figure, support the results. The Discussion section is well supported by their previous work in this field (stated in reference Nº 16), as well as by their current data.
Likewise, their Conclusions are in accordance with the interpretation of their results, giving also meaningful information about the study developed. Moreover, the authors explain the limitations and strengths of their study.
Noteworthy, the article offers a high quality of written English and a good coherence of the arguments.
However, to improve the manuscript, the following minor issues should be addressed:

Specific comments:
- Page 1, Line 24: Even though the abbreviation "AGEs" could be well known, please, clarify its meaning (i.e.: Advanced Glycation End Products) the first time it appears in the text, as well as in the keywords.
- Page 6, Line 180: Please, perhaps it would be better if Table 2, and its corresponding legend, appear jointly on the same page.

Therefore, the authors should revise the submitted manuscript, for addressing the aforementioned minor issues.

Author Response

Thank you for the positive support for the novelty, significance and limitations of the studies presented in this manuscript. Below is an itemised list of the changes we have made in accordance with your suggestions.

Reviewer comment:Page 1, Line 24: Even though the abbreviation "AGEs" could be well known, please, clarify its meaning (i.e.: Advanced Glycation End Products) the first time it appears in the text, as well as in the keywords.
Page 6, Line 180: Please, perhaps it would be better if Table 2, and its corresponding legend, appear jointly on the same page.

Response: We have included the full definition of the term AGEs (please refer to line 25) and have also adjusted all tables and figures including the one on page 6 to accommodate both the full table and the legend on the same page.

Please note, based on reviewer’s concern regarding the variance observed in the CML data we have removed this data from the manuscript.

Reviewer 3 Report

  1. Amelia et al studied the correlation of serum AGE and sRAGE levels with impaired glucose metabolism in T2D participants. The hypothesis is not new or no new information is obtained from the study.
  2. The author should include participant’s incretin hormones and glucagon levels and their correlation with circulating AGE and sRAGE to support their hypothesis on postprandial hormonal response
  3. No significant correlation was obtained between the control and T2D group in circulating levels of sRAGE and AGE levels. Including more participants with long diabetes duration will give some sensible correlations.
  4. The authors had mentioned not much difference in the BMI between the groups. But they had obtained a negative correlation with sRAGE and waist circumference. The authors should include the stimulated c-peptide or insulin data for the participants tested.
  5. The title is complex and not clear or catchy.

Author Response

We would, like to thank the reviewer for their careful consideration of the manuscript and constructive feedback of how to improve this work.  Below is an itemised list of our responses to your points. Please note, based on some reviewer’s concern regarding the variance observed in the CML, data we have removed this data from the manuscript.

1. Amelia et al studied the correlation of serum AGE and sRAGE levels with impaired glucose metabolism in T2D participants. The hypothesis is not new or no new information is obtained from the study.

We would politely disagree regarding your interpretation of the main findings of this manuscript and the novelty of this data. Although we agree that the relationship between the AGE/sRAGE axis with impaired glucose tolerance has been examined before, we were primarily interested in the relationship between glucose dosage and gastrointestinal handling and AGE and sRAGE levels in these uniquely matched participants. This was the primary aim of the manuscript as stated in lines 80-86 of the original version, (lines 72-79 of the amended version). Never before has the circulating receptor sRAGE been examined in the context of oral and intravenous glucose infusions at escalating glucose dosages within the same individual. This also provides unprecedented ability to examine sRAGE production after eating or following gluoconeogenic glucose production (intravenous delivery) akin to those seen physiologically and unpack the differences between these. We have taken the decision to remove the AGE data from the manuscript due to the variation in the data and we do not believe that it adds anything to the findings.

2. The author should include participant’s incretin hormones and glucagon levels and their correlation with circulating AGE and sRAGE to support their hypothesis on postprandial hormonal response.

Thank you for this interesting suggestion. We have now added these correlations to the manuscript describing the relationship between oral and intravenous sRAGE AUC and glucose, insulin, glucagon and the incretin hormones GLP-1 and GIP AUC (see table 3 and lines 185-188, and 200-203. We have not included baseline incretin data or the patients incretin responses as this data has already been published.

3. No significant correlation was obtained between the control and T2D group in circulating levels of sRAGE and AGE levels. Including more participants with long diabetes duration will give some sensible correlations.

Thank you, for this suggestion which we agree would be of interest.  However, this was not the purpose of these studies which are addressing acute sRAGE production in response to glucose variations and whether this differs between oral delivery and endogenous production as well as with type 2 diabetes. We also tested various glucose dosages representing low or high blood glucose concentrations. Some excellent studies showing longitudinal data on changes in plasma AGEs and sRAGE have been previously published and these are heavily dependent on factors such as longitudinal diabetes control. The novelty of our acute data within the same patient testing different glucose concentrations and sources (oral vs endogenous such as from the liver gluconeogenesis), as compared to previous studies.

4. The authors had mentioned not much difference in the BMI between the groups. But they had obtained a negative correlation with sRAGE and waist circumference. The authors should include the stimulated c-peptide or insulin data for the participants tested.

Matching was undertaken during recruitment of participants to ensure that the groups were matched for sex, age, and BMI. As such, there were no differences in BMI between groups (see Table 1). However, to clarify, we saw a trend towards significance only for a negative correlation between sRAGE and waist circumference and only in the combined group where both the controls and the patients with type 2 diabetes were combined into a single cohort (Table 1). As this was a trend only, we did not feel this warranted significant discussion, however the direction of the relationship observed was consistent with the previous studies we discuss in the discussion section. We believe some of the confusion regarding this matter may have arisen due to an erroneous title that was included in the results section 3.2. We have amended this title (see lines 136-137).

Thank you for your suggestion regarding the inclusion of insulin or c-peptide data, which are now presented in Table 1, please refer to page 4 of the amended manuscript.

5.The title is complex and not clear or catchy

We have tried to be as informative and clear with the title, whilst also including key words that will orientate readers to the findings of the study. However, we agree a clear and catch title can be helpful. We have shortened the title in accordance with our changes to the manuscript, however, below are some alternatives we would be happy to consider should the article be accepted. We would be happy to discuss this further with the journal’s editorial team.

Assessing the effects of increasing oral and intravenous glucose doses on soluble RAGE production in adults with and without type 2 diabetes mellitus.

The effect of increasing glucose excursions on soluble RAGE production in metabolically healthy individuals with type 2 diabetes.

Do glucose excursions and the gut affect soluble RAGE production in type 2 diabetes? 

Reviewer 4 Report

  • The manuscript entitled “Circulating levels of Age….type 2 diabetes” reports on the circulating levels of AGE carboxymethyllysine and sRAGE in response to increasing OGTT and IIGI tests. The manuscript is well written, wants to convey an important clinical message, and should be of great interest to the readers. However, the results are not adding any new information due to the limited sample size design of this study.
  • The authors report that in this study there is no difference in baseline levels of CML or sRAGE compared between control vs Type2 Diabetic patients contrasting the previous reports by (Kilhovd et al 1999 & Tan et al 2006). The major limitation of this study is the sample size which although being self-addressed by authors but introduces a major setback in this study. This is simply because a sample size that is too small increases the likelihood of a Type II error skewing the results, which likely is the reason for no difference observed in most of the results which potentially decreases the power of this study.
  • The same goes for other results/s where there is no CML increase observed with increasing glucose excursions with OGTT or IIGI.

Author Response

Please note that these were received only four days before the deadline for re-submission:

Additionally, please note, based on  other reviewers suggestion regarding the variance observed in the CML, data we have removed this data from the manuscript.

We would, like to thank the reviewer for their careful consideration of the manuscript and constructive feedback of how to improve this work.

  1. Power of the study: We acknowledge that this study was not specifically powered to measure sRAGE and we have deliberately highlighted this in the manuscript (see lines 270-273) acknowledging the potential for a type 2 error. This is a unique cohort, with the same subjects undergoing matched multiple OGTT and isoglycaemic glucose infusions, which allowed for subjects to be compared with themselves which maximises statistical power. It also has uniquely matched oral and intravenous glucose delivery where the blood glucoses are intricately matched at each timepoint. Both the repeated measures design and the rigorous matching between the cohorts adds significant extra power. This also provides unprecedented ability to examine sRAGE production after eating or following gluoconeogenic glucose production (intravenous delivery) akin to those seen physiologically and unpack the differences between these. As a consequence, we believe that we have found some interesting and significant findings.
  2. Baseline AGEs and sRAGE: Please note that we did not anticipate finding baseline differences between AGEs and sRAGE between the groups with diabetes and no diabetes. The reasons for this are (i) those with type 2 diabetes were only very recently diagnosed and treated with diet intervention only. (ii) all of the subjects were overweight (both groups, controls and patients with diabetes were considered overweight by clinical definition) and there are already some changes particularly in soluble RAGE reported in obesity (see Norata et al, below and Guclu et al below). We were however, aiming to see the differences that diabetes imparts on sRAGE in particular when glucose control has failed sufficiently for diagnosis of diabetes. We did not use subjects who were medicated nor who had very poor glycaemic control since we believe that would have been a more biased population and not necessarily representative. We acknowledge that clinical recommendations now include early use of metformin which is for arresting risk for complications (which HbA1c diabetes diagnosis is based on – i.e. when subject reaches 50% risk for retinopathy based on HbA1c).
  3. No acute changes in CML during OGTT/IIGI: We politely disagree with the reviewer and were not surprised by the lack of changes seen in CML. Given AGE chemistry, which can be extremely complex inside the body, it was unlikely that CML could be modified in such an acute manner. Indeed, an excellent previous study has shown changes in precursors of AGEs such as 2-deoxyglucasone and methyglyoxal which definitely change acutely during a 75g oral glucose tolerance test (see reference below by Maessen et al). It is important to note, that if samples are not treated with extreme care, that glucose and other AGE precursor present in diabetic plasma can freely from AGEs “ex vivo” which is common in individuals with diabetes given their blood glucose concentrations. We have tried to take great care in handling of the samples which allows us to suggest that there is not acute formation of CML in response to blood glucose concentrations, but does not preclude the formation of other AGE precursors nor of other AGEs such as MG-H1. However, given we did not see any differences and the variation in the CML data and the impact this has on the possibility for a type 2 statistical error, we have removed the CML data.

Maessen, D.E., et al., Post-Glucose Load Plasma alpha-Dicarbonyl Concentrations Are Increased in Individuals With Impaired Glucose Metabolism and Type 2 Diabetes: The CODAM Study. Diabetes Care, 2015. 38(5): p. 913-20.

Norata GD, Garlaschelli K, Grigore L, et al. Circulating soluble receptor for advanced glycation end products is inversely associated with body mass index and waist/hip ratio in the general population. Nutr Metab Cardiovasc Dis. 2009;19(2):129-134. doi:10.1016/j.numecd.2008.03.004

Guclu M, Ali A, Eroglu DU, Büyükuysal SO, Cander S, Ocak N. Serum Levels of sRAGE Are Associated with Body Measurements, but Not Glycemic Parameters in Patients with Prediabetes. Metab Syndr Relat Disord. 2016;14(1):33-39. doi:10.1089/met.2015.0078

Round 2

Reviewer 3 Report

I thank the authors for carefully addressing all the queries.